# From Insight into Action: Understanding How Employer Perspectives Shape Endometriosis-Inclusive Workplace Policies

**DOI:** 10.3390/healthcare13080930

**Published:** 2025-04-18

**Authors:** Danielle Howe, Michelle O’Shea, Sarah Duffy, Mike Armour

**Affiliations:** 1NICM Health Research Institute, Western Sydney University, Sydney 2145, Australia; d.howe2@westernsydney.edu.au (D.H.); m.oshea@westernsydney.edu.au (M.O.); 2School of Business, Western Sydney University, Sydney 2116, Australia; sarah.duffy@westernsydney.edu.au; 3Medical Research Institute of New Zealand (MRINZ), Wellington 6021, New Zealand; 4Translational Health Research Institute (THRI), School of Medicine, Western Sydney University, Sydney 2145, Australia

**Keywords:** endometriosis, menstruation, menopause, policy, guideline, work, employers, gender mainstreaming, occupational health and safety

## Abstract

**Background**: Endometriosis costs Australia $9.7 billion AUD annually, with absenteeism and lost productivity mostly contributing to this cost burden. Previous research has shown how the absence of workplace support(s) for endometriosis symptom management can exacerbate negative effects. Despite this knowledge, the role of employers and workplace policies in supporting employees with endometriosis remains unexamined. **Background/Objectives**: As part of the Endo@Work project (co-designed endometriosis workplace guidelines), this study examines the perceptions and experiences of managers, HR, and senior leaders to understand how the Endo@Work guidelines can be successfully implemented. **Methods**: Seven focus groups with 24 Australian employers were undertaken. Using reflexive thematic analysis, data were interpreted through a post-structural feminist lens with focus group discussions revealing how employers recognise the importance of workplace guidance/policies. **Results/Conclusions**: Problematically, existing policies and practices were often tokenistic and inconsistently implemented. This study identifies how leadership buy-in, flexible work arrangements, and the thoughtful integration of workplace education initiatives contribute to supporting employees managing endometriosis symptoms at work. This study’s findings emphasise how consistent policy enactment alongside ongoing education/training foster inclusive workplaces and gender equity outcomes.

## 1. Introduction

As women’s labour market participation inches closer to parity [1,2], scholars have sought to understand women’s work and career experiences. The gender pay gap, workplace sexual harassment, and women’s leadership experiences [3,4,5,6,7,8] are increasingly examined. An emerging body of research is critically addressing how work cultures and practices ‘discipline’ women’s bodies [9,10,11] and inequitably shape their career trajectories [12,13,14]. The effects of these disciplinary practices are in part associated with workplace values which reproduce the “ideal worker” norm. The ideal worker is typically an unencumbered employee who prioritises work above personal, familial, or social obligations [12,13,15].

Relatedly, and tied to this ideal worker norm, there is a growing recognition of how women’s working lives are shaped by menstruation, menopause, and chronic illnesses, including endometriosis and chronic pelvic pain (CPP) [14,16]. Employees who “fail” to embody this ideal include parents, people with chronic illnesses, women, and people presumed female at birth (PFAB) who menstruate, experience menstrual disorders, and transition through menopause—as their bodily functions are often considered productivity disruptions. Furthermore, conditions such as endometriosis, CPP, problematic menstruation, and menopause symptoms are not meaningfully or sensitively addressed in the workplace in part because of the shame and stigma associated with reproductive health [17]. Relevant to the present research, women and those PFAB living and working with problematic endometriosis symptoms often fail to embody the ideal worker, with the unpredictability and severity of their symptoms reducing their perceived ‘commitment’ to work. Accordingly, the salience of the ideal worker norm reinforces narrow definitions of a ‘good worker’—one deemed worthy of reward and promotion [12].

This paper sometimes uses the term ‘women’ and ‘women’s health’. We do so in order to acknowledge the specific experiences and challenges faced by cisgender women at work, often exacerbated by menstruation, menopause and/or chronic illnesses including endometriosis and chronic pelvic pain. Additionally, our language use reflects the cisgender cohorts of women prior research draws on. The authors of this paper acknowledge that trans men, intersex, non-binary, and gender-diverse people can menstruate, transition through menopause, and/or have chronic illnesses, including endometriosis and chronic pelvic pain. By using the term ‘women’, this paper both accurately reports the cohort represented in existing studies, while also illuminating the gaps in our understanding of the specific experiences, challenges, and supports for gender-diverse people at work. Where possible, this paper uses both terms—women and those presumed female at birth (PFAB), as both groups may be impacted by endometriosis. Our language choice is underpinned by research and guidance on inclusive language and accurate reporting [18].

To advance workplace gender equity outcomes, occupational health and safety approaches that value and enable women to manage their gynaecological health at work are increasingly being prioritised [14,19,20,21]. For example, endometriosis, a chronic inflammatory condition that affects 10–14% of women and those PFAB [22,23,24]. The condition can impact many, if not all, aspects of a person’s life, including their work and career [25,26,27,28]. The cost of illness burden for endometriosis is estimated at 9.7B AUD per year, with the majority of costs (84%) attributed to lost workplace productivity [29].

Endometriosis negatively impacts productivity, absenteeism, career choices and progression, and professional life [25,27,30,31]. Debilitating endometriosis symptoms, combined with limited work flexibility and support, contribute to presenteeism (i.e., working with reduced productivity) and absenteeism (i.e., missing work) [32]. Women diagnosed with endometriosis are more likely to transition out of the labour market when compared with women without the condition [22]. Those who suffer more severe symptoms report lower workability [33] and a higher probability of unemployment [16]. Accordingly, to advance gender equity outcomes, the development and implementation of workplace guidance associated with chronic health conditions such as endometriosis and chronic pelvic pain is essential.

Research suggests that relatively small workplace adjustments can improve endometriosis symptom management and work performance, as evidenced during the COVID-19 pandemic and associated lockdowns forcing employees to work from home. Adjustments including but not limited to flexible work schedules, adequate breaks, modified physical loads, access to nearby bathrooms, and the ability to rest or use heat packs as needed were reported to greatly assist employees in managing symptoms and improving productivity [34,35]. Despite these gains, there has been little empirical research providing employers with guidance on how best to develop and implement supportive policies and practices post-pandemic for people with endometriosis [36]. As workplaces fundamentally changed during the pandemic, this research attempts to address this knowledge gap, taking place entirely in ‘post-pandemic’ Australia.

This study is the fourth phase of a broader project (titled: Endo@Work) involving the co-design of a set of evidence-based guidelines for Australian workplaces. The first phase of the research involved a critical global scoping review of workplace policies, guidelines, and legislation to support menstruation, menstrual disorders, and menopause at work [36]. The review identified the fractured approach to addressing varying aspects of women’s health at work—highlighting the lack of evidence underpinning policy design and lack of evaluation to understand efficacy or impact [36]. Phases two and three sought to address this gap through a participatory approach, centralising and valuing the voices and experiences of Australians living with endometriosis to co-design a set of workplace guidelines. Phases two and three included a national online survey generating a snapshot of peoples’ experience with existing policy and managing endometriosis at work [37]; and six focus groups with 45 participants (recruited from the survey) exploring women’s experiences managing endometriosis at work [38]. These data (phases 2 and 3) informed a set of workplace guidelines consisting of three pillars—policy, processes, and promotion—with the aim of supporting people with endometriosis at work. Missing, however, were the voices and experiences of Australian employers (e.g., managers, supervisors, and HR specialists) who can provide practical insight and expertise to co-design and refine the Endo@Work guidelines, ensuring their relevance and applicability to Australian workplaces.

This paper explores the perceptions and experiences of employers in various roles, including managers, supervisors, senior leadership, and human resources (HR) representatives, in managing the work of employees with endometriosis or similar health conditions. Accordingly, the present research aims to (a) provide practical guidance for employers seeking to support employees managing endometriosis symptoms at work, and (b) recommend how guidance and supports can be integrated across existing workplace policies and practices. The study is guided by the following two research questions:

1.What workplace policies and supports currently exist, and how do they compare to the proposed Endo@Work guidelines?2.How have existing workplace interventions and/or supports enabled employees with endometriosis to work productively while managing endometriosis symptoms?

## 2. Methods

### 2.1. Participants and Recruitment

After receiving the ethics approval from Western Sydney University (H15537), this study (focus groups) was advertised through social media platforms, including Instagram via Endometriosis Australia, an endometriosis advocacy organisation, and associated online platforms supporting the endometriosis community (>45,000 combined followers in 2022). The research team also shared the recruitment invitation through their personal and professional networks. This study was open to people living and working in Australia who are over the age of 18 years and who are currently employed in a managerial, supervisory, executive, and/or HR role. Participants self-selected to follow a QR code on the study’s advertisement to complete an expression of interest form, via the online survey platform Qualtrics. Contact and demographic information including job title, organisation name, industry/field, and number of employees were collected. The primary author (DH) contacted via email all the respondents who met the inclusion criteria and assigned people to focus groups based on their availability.

This project adopts a participatory action research (PAR) approach by emphasising a continuous commitment to understanding participants’ lived experiences [39]. Extending Foucault’s “power–knowledge relations” [40], the study approach is embedded in the embodied experiences, perceptions, and understandings of its participants [41].

Between October and November 2023, seven focus groups were conducted, involving a total of 24 participants. Each session lasted one hour and comprised three to four participants, accompanied by two authors. All the sessions were facilitated by DH, with MO, SD, and MA each attending at least one session. The majority (92%) of the participants identified as cisgender women, with the remaining 8% identifying as cisgender men. Most participants were employed in medium-sized organisations (25%, with 50–199 employees) or large organisations (50%, with over 200 employees). Approximately one-third worked in national workplaces, while Victoria (42%) and New South Wales (29%) were the most represented locations. Participants were employed in a diverse range of industries, offering varied perspectives across different workplace types (See Table 1).

### 2.2. Procedure

The participants engaged in a 60–90 min online focus group (via Zoom). The focus groups followed a semi-structured discussion informed by preliminary findings (phases one and two), and acted as open-ended prompts to explore the following:Experiences managing people with endometriosis (and similar conditions) in the workplace;Experiences with workplace accommodations, supports, and policy;Perceptions and feedback on the research team’s co-designed Endo@Work guidelines (see next section Reviewing Endo@Work Guidelines).

Following each focus group, the researchers leading the session stayed online to discuss their initial observations and reactions. Reflexivity, an important facet to feminist qualitative research [42], was integral to this phase. During this stage, the researchers refined the open-ended prompts and questions in preparation for the following session.

### 2.3. Reviewing Endo@Work Guidelines

Before the focus groups, the participants were asked to review a draft version of endometriosis guidelines (Endo@Work), including policy recommendations, educational factsheets, and processes/promotion recommendations. All the guidance/recommendations included key findings and best practices informed by previous phases of the Endo@Work research [36,38].

The guidelines reflected three pillars: policy, processes, and promotion. Regarding ‘policy supports’, flexible working arrangements and the ability to take breaks and/or access the toilet and access a quiet break room dominated focus group discussions. Concerning sick leave and formal requests for accommodations, a one-time medical certificate was recommended. Involving ‘processes’, the second pillar included prescriptive steps for an organisation to take to successfully embed policy into ongoing processes. For instance, instructions on embedding reasonable adjustments for staff, roles, and responsibilities for a workplace “EndoChampion” [38]. The EndoChampion (also discussed in this study’s themes) was introduced in the Endo@Work guidelines to serve as a supportive and approachable workplace advocate and knowledge resource. The nominated ‘champion’ does not need to have an endometriosis diagnosis. Empathy, highly developed communication skills, and a willingness to learn about the condition dominated discussions. EndoChampions were described as helping foster inclusivity and direct colleagues to appropriate resources. The third pillar, ‘promotion’, included education in the form of supervisor and employee factsheets and how the nominated ‘champion’ can practically promote endometriosis awareness in the workplace. Finally, participants discussed their experiences implementing similar guidelines or supports, experiences, and/or perceived challenges.

### 2.4. Data Analyses Approach

Reflexive thematic analysis (RTA) [43] was employed to explore how endometriosis symptom management in the workplace was experienced and how these experiences intersect with broader societal systems and structures. By drawing on Braun and Clarke’s (2006) framework, RTA offers a structured method for identifying, organising, and capturing patterns of meaning within qualitative data. Familiarising oneself with the data, generating initial codes, identifying latent themes by clustering conceptually similar manifest codes, refining themes and assigning representative labels, and producing a detailed report that highlights the relationships between themes underpin their approach [44]. The analysis was inductive, with the data driving the conceptualising of themes [43,44]. During the initial analytic phase, the primary author (DH) de-identified all the transcripts, assigned pseudonyms to participants, and compiled them alongside authors’ initial reflections into Word documents. To deepen data familiarity, DH repeatedly reviewed the de-identified transcripts and accompanying notes, documenting observations and reactions. She then engaged with all the authors to discuss these initial reflections.

Through the second phase, DH developed preliminary codes by re-reading transcripts and organising the data alongside their corresponding codes in a structured Word document. A table was used to list the initial codes in the left column, while the related line-by-line transcript data were assigned to the appropriate code. In the third phase, DH collaborated with the authors MA and MO to review and refine the initial codes. Together, they grouped codes based on conceptual similarities and identified latent themes to uncover deeper meanings. Reflexivity was maintained by acknowledging assumptions, resolving coding discrepancies, and collectively reaching consensus on latent themes. Finally, to cross-check codes and themes, DH used the qualitative data analysis software NVivo 12.6.1 to systematically recode the transcripts line-by-line under the agreed-upon latent themes. All the quotes are presented using pseudonyms.

### 2.5. Positioning Ourselves in Data Analyses Approach

This study adopts a post-structural feminist lens to understand the experiences of employers (managers, supervisors, and HR specialists) managing employees with endometriosis. The focus groups were facilitated using a feminist approach, emphasising sensitivity, empathy, and engagement with participants’ lived experiences. Rather than maintaining a detached, impartial stance, the facilitator actively acknowledged and responded to participants’ insights, fostering a collaborative and supportive environment. This approach aimed to validate participants’ perspectives, encourage open dialogue, and recognise the co-construction of knowledge within the research process. Post-structuralism offers a framework that acknowledges the diversity of knowledge, truth, rationality, and power [45,46]. By extending “problematisation” as a method to ‘stand back’ from ‘objects’ and ‘subjects’ presumed to be objective and unchanging (e.g., workplace structures: policy and processes) to look “at the unfolding, the evolution and the interaction of different practices” [47], we can usefully contribute to alternative ways of workplace organising [48] that enable the involvement and productivity of a diverse workforce.

## 3. Results

In principle, most managers suggested that policy, guidelines, and reasonable adjustments were valuable approaches to supporting employees managing problematic endometriosis symptoms at work. Despite this in-principle support, there was little knowledge or consensus on how to design inclusive policies. Theme 1 illustrates employers’ perceptions that existing policies were tokenistic, inconsistent, and were inequitably applied. Theme 2 illuminates the everyday practices (via top-down, trust, and advocating) shaping inclusive workplaces (Table 2).

### 3.1. “We Have Policies”

#### 3.1.1. We Have Policies…but They Are Failing to Fully Support All Staff

Most participants stated that their workplace offers a policy relevant to flexible working arrangements and/or flexible practices as outlined through the Endo@Work guidelines. Concerning menstruation (leaky bodies at work), none of the participating workplaces had initiated a menstrual or menopause policy/guidance. Accordingly, while participants reported that policies existed for flexibility, the unspoken norm that you ‘just get on’ with the job was strengthened.

For many employers, existing national standards informed workplace policy developments. Sam made the following remark:


*“…there’s the national employment standards that organisations have to meet in terms of their obligations around offering flex [flexible work practices]”*
Sam

The nature of the industry/work also shaped flexibility outcomes. Jordan explained the following:


*“In the manufacturing industry…we would have about 80% blue collar workers and … 20% white collar. The recommendations that you’d put in [Endo@Work], all of those are either already implemented or could easily be implemented. But…when I reflect…at previous workplaces that are probably more predominantly white collar, 100%, they’re probably already introduced”.*
Jordan

Sam outlined (FG1) how the business/workforce size impacts formalised policies:


*“…most [workplaces] will have a flexible work approach or policy, particularly in the larger organisations, often not in the smaller ones, but they’ll be a bit more ad hoc in how they approach flex”.*
Sam

However, when the research team asked respondents to discuss *how* staff were accessing the policies, several issues emerged. Foremost among these issues was their informal and inconsistent application, such that they were described as a ‘tick-a-box’ exercise. *“Looking at our policies… how do we make it consistent? We could be doing better”.* Grace

Further problematically, policies were either not promoted, or different teams/organisational departments identified challenges accessing the policy:


*“It tends to be a bit tokenistic…there’s a lack of promotion because…it’s sometimes easier to just kind of go, yeah, it exists and we’re doing the right things as far as we’re keeping our noses clean from a legislative and regulatory perspective, but we’re not going to encourage our people to use it and we’re not going to tell them how they can use it, when…and why they can use it…that sort of process piece is absent”*
Olivia

Without policy advocacy from management at all levels, employees may not feel comfortable to take up the supports offered, as follows:


*“We support people, we just don’t loudly support them. That doesn’t allow for an environment where people who are suffering from endometriosis or any other chronic illness feel safe to come up and say, hey, this is what I’m suffering with. This is my pain management; this is what I might need”*
Grace


*“We know that we’re a supportive firm and that we would be 100% flexible with any team member, with whatever their needs may be. But how do we make sure that each employee and each team member feels and knows that as well? I’m starting to think now, is anyone being left behind?”*
Natalie

These responses illustrate how the ideal worker norm persists and is strengthened, further highlighting the importance of workplace culture in shaping daily practices, despite formal policy countering ideal worker norms.

#### 3.1.2. We Have Policies…but They Are Inconsistent Across Managers

In most workplaces, line managers were positioned as policy gatekeepers who could render policy access easy or difficult. Matt remarked how “*…every individual line manager would deal with it differently*”. Other participants discussed how access to support was influenced by the line managers’ experience living with endometriosis and management style, as follows:


*“I think biases play a huge part in this, and I know that it shouldn’t, but it does. As someone who’s experienced endometriosis for a long time, I will do whatever I can to keep [flexibility] ‘off the books’, to help somebody within the realms of appropriateness…But for other managers, it would just be, ‘absolutely not. We’re going to go by the letter of the law and we’re going to follow due process.’ So, I think it really comes down to the individual”.*
Olivia

Sam also remarked how


*“One of my staff couldn’t figure out for quite a while why she was in a lot of pain. Then she found out that she has endo. I had quite a negative experience with my employer at the time because they were not open to allowing her to work from home. So…I just said, I know she’s a hard worker. I know that she’s not taking me for a ride. I told her, you do what you need to do. She does an incredible job. So, it was really hard…the senior management was predominantly male. They don’t know anything about endo. There was very little want to know and to understand that it’s chronic illness and what it entails. So, there was very little support for me. And at some point when I put her forward for [a promotion], I was told that, well, she needs to be aware that she needs to come into the office and the fact that she wants to work from home is not going to work out…I had… [to] champion for her, but it’s not very common and there is no policies and there is very little support to understand that it’s a chronic illness and that the work can be done”*
Sam

Workplace leaders play a crucial role in the success or failure of policy interventions.

#### 3.1.3. We Have Policies…but They Are Only Beneficial to the Ideal Worker

In a post-pandemic work setting, many workplaces have implemented a company-wide or one-size-fits-all ‘flexibility’ policy. Problematically, this approach is wedded to ‘hours worked’ rather than work outcomes and/or outputs. Our data demonstrate that this one-size-fits-all approach may be ineffective in supporting people with endometriosis because endometriosis is a chronic (non-cyclical) condition whose symptoms vary from person to person; therefore, types of support vary. Matt elaborated the following:


*“I think you’ve got to define what flexibility is a little bit. Is flexibility working from home? Is it being able to work from home sometimes? Is it flexibility within an office environment? And what does someone actually see [flexibility] as being? Because I know when we, as an example, we came back from COVID I thought that we needed to be in the office a little bit. So, we went three days in the office, two days out of the office. Then after three months, I asked the team, ‘do you think this is working?’ And it was, ‘no’. So, I suggested, ‘why don’t we default [everyday] working in the office? And if you’ve got to work from home, then work from home’.…That was more flexible than when we had certain days in the office”*
Matt

Workplaces adhering to strict expectations around *when* employees’ are permitted to work flexibly may fail to support staff with chronic illnesses, such as endometriosis, where flare-ups are non-cyclical and can vary, for example:


*“What people need at different times is very different. You might have someone who’s very acutely unwell, having a lot of surgery, and then they might have a period of being well for a year or 18 months. It’s really hard to…spell out people’s experiences because everyone’s [symptoms] are just so different and what support they need, it will differ at different times. Having that flexibility and responsiveness and openness and to review these actions…is probably more beneficial”.*
Georgia

The physical office lay-out was also problematic for some staff managing endometriosis, as follows:


*“There’s a lack of privacy [hot desking], especially if you cannot keep your personal items at your desk. So, if you need to bring into work your heat packs, with you every single day because you’re working at a hot desk, that’s not ideal for people with chronic illness…And then just comfortable seating, like leg rests and things. Some of the seating we have is uncomfortable, so being able to bring in a cushion, wearing comfortable clothes to work, we all dress very corporate in my office, which is fine, but when you’re having a flare-up, you just want to wear something that’s a bit more comfortable. So these are just like really simple examples of: can the workplace be accommodating in small ways”*
Mia

Relatedly, the physical infrastructure available in some workplaces presented obstacles for employees seeking to manage symptoms while physically present in the workplace, as follows:


*“So … we’re in a good space prior to this, though, and a lot of my colleagues, female colleagues are experiencing this, is that we get deployed to sometimes quite different locations where you don’t have access to the type of toilet or microwaves, heat packs, all the comfort items. And there is an expectation in my job that we get on with it and you just have to deal with it”.*
Riley

For meaningful change to occur, all elements of the workplace need to be evaluated, including but not limited to flexibility policies, uniforms or dress standards, office infrastructure, and even if employees are expected to work a certain number of hours or achieve particular outcomes.

### 3.2. Workplace Culture Change Is Driven by the Banal (Everyday) Practises

Our data illuminate how policy socialisation arises through informal everyday practices. While the introduction of policy is a key step, positive change to support people with endometriosis is driven by everyday managerial and leadership-led practices.

#### 3.2.1. Change Is Driven from the Top-Down

Most study participants acknowledged that culture (change) is driven from the top-down. For example, Jane provides the following case where a manager who does not apply strict rules and norms around work can help drive an inclusive and supportive environment:


*“The people that were managing the centre before…were very stringent. No food at desks, signs everywhere, and just telling people how to be human beings. And it was just not a nice environment…A few people felt uncomfortable. They went to HR. HR…realised there was a fundamental problem with the way that the [workplace] was being managed. So those managers were moved on [and] a new manager was brought in. He was tasked with really shifting the culture…it’s just appreciating the people that we have and just trying to make it as inclusive and supportive and comfortable as we can”.*
Jane

During the discussion, another participant, Jen, elaborated on the importance of senior leadership buy-in in enabling a top-down cultural change, as follows:


*“…it is that cultural thing from the top…if the top of your organisation is advocating for flexible work, open and honest conversations, then it flows down to the next level of management, and then it flows down to my level of management, which is middle management, which is the people on the ground who can say, ‘yeah, look, I’m happy to accommodate your flexible working request within operational reasons’”*
Jen

Even in male-dominated workforces (numerical and cultural), where ideal worker norms remain deeply entrenched, Margaret presented the following case for how senior leadership contributes to cultural change:


*“…if we send an internal communication out via email. The way that they think about the information that’s in their email is how much money worth of time is it going to take for our entire business to read that? If everyone’s average salary is $250,000 a year and we’re asking them to dedicate a portion of their time at work to read that email, it better be worth it. So, it is a challenge, but if I can…plant the seed among some of our senior leaders…[and] keeping them mindful that this is a problem, and it is something that people struggle with every single day”*
 Margaret

Implementing policy or cultural change through a top-down approach can foster more consistent adoption across the workplace. Lydia made the following remark:


*“It needs to be modelled…by all of management so then people feel safe to do it. And that’s where, in our organisation, it is modelled by our senior leaders that you do work four days a week, that you’re not expected to answer your emails when you’re not there or your phone”.*
Lydia

Positional power and seniority within the workplace can create workplace cultural change or maintain the status quo.

#### 3.2.2. Change Is Driven by Managers Who Trust Staff and Understand They’re Not Homogenous

When prompted, participants in the focus groups provided examples of how they have successfully supported their staff with endometriosis (or similar issues). Many participants reported the importance of understanding how *all* staff have different needs and require varying levels of support. They suggested that meaningfully meeting these needs is tied to establishing and maintaining trust. For example, participants interpreted how employees are experts of their health, are best placed to communicate when they require additional support, and that most employees, if supported, will be productive at work. Matt made the following remark:


*“[Managers need to be] willing to say… ‘I’m going to be led by you and I’m happy to be led by you as much as possible’. And a lot of that comes down to trust the person that’s working for you. If they get it wrong, then deal with that, but go with the view that they want to do a good job and you’re just providing an environment and the tools for them to be able to do that rather than coming at it as, ‘well, that person’s going to need this. So actually, they’re just having-me-on’…”*
Matt

Natalie also reflected on the positive effects of trust, as follows:


*“I’ve got team members who have endometriosis, and for the firm that I work in, we have quite a large female or assigned a female at birth team, so we’re quite female leading…A lot of the things in the [Endo@Work] guidelines are already in practice for us. We have quite a trust-based culture…we are always giving our team the time that they need to manage any of their health or life responsibilities”*
Natalie

Tied to the discussion of trust, Lydia interpreted how transactional relationships with staff can undermine employee commitment and work outcomes, as follows:


*“So, it’s…humanising it, sitting with a person and just going, I don’t really understand what you’re going through, but I’m here for you. And you’ve got a real challenge in that because you work in a workforce that is so…transactional, right? Yeah, but I know the power of having that within an organisation…with people feeling acknowledged with their struggles, whether that be physical or mental health. It’s trying to find that thing, that point of sameness or something…getting through to somebody somehow”.*
Lydia

The success of more inclusive and supportive workplace policies requires a high degree of belief and trust in employees following through on their commitments.

#### 3.2.3. Change Is Driven by Consistent Education and Advocacy

All seven focus groups recognised that normalising endometriosis (and women’s health more broadly) was an important factor in creating culture change and promoting policy/guidelines. “*If the conversations are normalised around women’s health in general, which is very different to men’s health, then…that creates a far better environment*” Matt.

Participants perceived that there were two successful approaches to advancing endometriosis (and health promotion)—consistent education and advocacy, as follows:


*“I honestly think for most in large organisations, it tends to fall on the outspoken few…it feels like [policy] is more tokenistic than anything else. At an organisational level…we throw some tampons and pads in the women’s restrooms and that’s it. We’ve got something visual that says, ‘look at us!!’ But it really does come down to those people that are willing to kind of be outspoken and…stomp their feet a little bit to lead the way for others, which is a bit of a shame in this day and age, but we’re making headway”*
Georgia

Most participants responded positively to the creation of an EndoChampion role, as proposed in the Endo@Work guidelines. The role intends to foster inclusivity by promoting education and direct colleagues to appropriate policy supports and resources in the workplace’s endometriosis guidelines. Through this role, workplaces could embed consistent education (informed by lived experience). Most participants thought the process could be embedded with relative ease: “*That would be…[an] easy win. Like…low-hanging fruit, because you’ve got…mental health first aid, as mental health champions, you’ve got LGBTQ…I think a lot of organisations would be open to it*”. Jordan.

Olivia referenced earlier discussions associated with ‘not expecting line managers to know everything’. The EndoChampion would enable consistent education across the organisation without having too great an expectation of line managers, as follows:


*“I honestly think my career has been mainly large-scale organisations, so hundreds, thousands of employees. I think the challenge is always making sure that it’s not a discrimination or psychosocial hazard checkbox because that tends to happen. I think the role of the champion is…critical here. We’re expanding our policies and leave, and the flexibility that we’re providing for employees around mental health, domestic violence, menstruation, menopause, and endometriosis and I think that’s wonderful. But you can’t expect line managers to have a good understanding of all those things, nor be comfortable having the discussion. Having Champions that are well educated… trained, that understand the difference between providing advice versus support, and the workplace does a really good job of communicating who those champions are”.*
Olivia

The stigma associated with endometriosis and CPP still lingers and will not be overcome without education and open communication with all staff.

## 4. Discussion

Guided by the research questions, this study examined employers’ experiences in managing employees with endometriosis or similar conditions. It also assessed existing policies and practices, comparing them to the proposed Endo@Work guidelines. This study presents two key findings. First, post-pandemic (in)flexibility still rewards the ideal worker—an unencumbered employee who prioritises work above personal, familial, or social obligations. This is evident from the workplaces reporting pre-existing policies and practices aimed at supporting employee health and wellbeing, which this study found were often tokenistic, inconsistent, and inequitably applied across employee cohorts. Second, policy alone is not enough and must be accompanied by top-down education, trust, and advocacy (often embedded through a support role, such as an EndoChampion). Specifically, this approach can contribute to challenging everyday (banal) practices and help to transform workplace cultures responsive to the diverse needs of the workforce towards greater equity and inclusion. Lastly, this study recommends workplaces take a top-down approach to embed tangible flexibility and deliver education and embed advocacy alongside policy, as practical guidance to support employees with endometriosis.

### 4.1. Post-COVID (In)Flexibility Still Rewards the Ideal Worker

The COVID-19 pandemic fundamentally changed how workplaces operate. As illustrated through the study findings, most workplaces have shifted to a ‘new normal’ and introduced new working from home and flexible working conditions [49]. Despite some moves to mandate that employee’s return to the office, a scoping review exploring the impacts of the pandemic on workplace productivity revealed that managers and employees adapted well to remote and flexible working, achieving productivity levels similar to those before COVID-19 [50]. The review findings also align with participants’ discussions of how genuinely flexible work practices—which enable employees to manage their endometriosis symptoms—can contribute to positive outcomes for individuals and institutions. The fact that hybrid working continues to benefit both employees and employers proves that deeply entrenched workplace norms (such as the expectation that an ideal worker must be in the office for eight hours a day, five days a week) can be challenged and transformed into a mutually beneficial ‘new normal’ [13,49].

Despite the envisaged mutually beneficial outcomes, the findings of this study suggest that changes made so far (post-pandemic) to workplace policies and practices are still enveloped by ideal worker norms that discipline women’s bodies at work and shape gendered and other intersecting workplace subjectivities [51]. For instance, employers still set expectations around flexibility in assigning days that staff must be in the office or can work from home (e.g., assigning Wednesdays as a work-from-home day). Placing this ‘norm around flexibility does not benefit a person with a chronic and unpredictable condition, such as endometriosis, as they cannot ‘schedule’ a ‘flare-up’ or fatigue to coincide with their workplace’s assigned work-from-home days. This is also generally the case for menstruation and menopausal bodies more generally. The unpredictable nature of heavy menstrual flow, chronic pelvic pain, and menopausal symptoms [52,53] still persistently prevents people from embodying this unencumbered ideal worker whom would benefit from these post-pandemic (in)flexible working norms [12]. This study shows that this only serves to perpetuate a norm that menstruating women/PFAB are to ‘just get on with it’, compelling them to hide and conceal their menstruating or menopausal bodies to reflect a socially acceptable ‘professional self’ to succeed in the workplace [10,11].

If workplaces wish to be inclusive and enable genuine flexibility, then they need to reimagine workplaces, and recognise that regardless of gender, all staff are likely to have *something* which takes precedence over work [12]. Research calls for workplaces to enact tangible flexibility [37,38]. Tangible flexibility refers to concrete, practical measures that allow employees to adjust when, where, or how they work to accommodate their individual needs without compromising organisational goals [54]. Unlike abstract policies or general principles, tangible flexibility focuses on actionable strategies that directly impact employees’ work–life balance and productivity. Examples include (but are not limited to) individualised flexible work hours, remote work options, job sharing, and adaptable workspaces. Previous research demonstrates that employees with endometriosis who had access to tangible flexibility took significantly fewer sick days and were significantly associated with improved presenteeism [37]. Tangible one-on-one flexibility would enable workplaces to better support people with endometriosis, as well as others who struggle to embody the ideal worker. This includes people who menstruate, transition through menopause, people with caring responsibilities, and people with chronic illnesses and disabilities. In turn, this inclusive flexibility may improve productivity levels establishing a mutually beneficial ‘new normal’ [50].

### 4.2. Policy Alone Is Not Enough

The findings demonstrate how policy frameworks alone cannot sufficiently challenge inequitable dynamics and form inclusive workplaces. Designing an inclusive policy is an important first step; however, it will be ineffective if the organisation does nothing to address the (banal) everyday practices reproducing and strengthening gendered and intersecting power relations [51,55]. For instance, a policy may offer tangible flexibility for people with endometriosis, as recommended above. However, if employees who use the policy face negative treatment, are perceived as ‘lazy’, or are overlooked for opportunities and promotions, they may avoid using it out of fear of discrimination.

This paper argues that policy as a ‘solution’ to inclusive workplaces must also disrupt the norms perpetuating inequality regimes [51]. Leading from Foucauldian post-structural feminist ideas, managers enact informal everyday practices which discipline and punish employees’ sense of self and their subjectivities [55,56]. Accordingly, we advocate for policies to be accompanied by education for supervisors and managers that provide advice and encouragement about how to “disrupt the relations of power that constitute the ‘ideal’ worker as masculine” [57]. For instance, our findings provide multiple cases where managers (who understood how endometriosis impacts women at work) were able to create a cultural change through everyday (banal) practices that helped women with endometriosis feel supported. Key supports included trusting employees to get their work done (regardless of if physically in the office or not), listening to employees and respecting their needs, advocating to prioritise their health needs, and being transparent about accessing flexible workplace supports themselves to create an open culture.

Research has demonstrated how a top-down approach with a shared strategy across leadership can incrementally drive culture change, which has been shown to enhance economic performance and the quality of working life [58]. Additionally, it is advantageous if the education (complementing policy) is informed by lived experience. This approach to education and training (or unlearning) helps to challenge the ideal worker norm and address the mechanisms and day-to-day processes that (re)produce to systemically discriminate against people with endometriosis based on their gender and ability [38]. Different workplaces will require tailored (un)learning resources, since the roots of inequality differ across industries (future research focused on specific sectors could help create targeted educational tools) [59]. However, the success of these efforts will depend on commitment among senior leadership. Introducing a workplace advocate role, such as an EndoChampion, who also has a direct line to senior leadership, can ensure that the strategy/education is effectively and consistently supported across leadership and the workforce [60]. This role could be adopted by existing diversity and inclusion offices/officers, human resources, and/or lived-experience roles [60,61]. For any solution to be effective, it will require input from a diverse range of stakeholders, including occupation health and safety representatives, human resources, senior leadership, and operations.

### 4.3. Some Key Recommendations Have Come from This Work

The following recommendations reflect discussions reported in Theme 2, where employers shared insights on potentially effective supports, approaches, and processes for employees managing endometriosis symptoms at work:Tangible flexibility: workplace policies that enact tangible flexibility offering individualised support plans (within the agreed-to terms in the policy);Create Guidelines that deliver education alongside policies: while policy is an important step to enacting change, education (informed by lived experience) is instrumental in normalising endometriosis, redressing stigma, and reducing associated discrimination;Advocacy role: a workplace advocate, like an EndoChampion, with direct senior leadership access can help ensure consistent support for strategy and education, potentially absorbed within diversity, HR, or lived-experience roles;Top-down approach: workplaces need to ensure leadership buy-in and advocacy through a stronger educational/training focus for senior leadership, supervisors, and managers for a top-down approach.

### 4.4. Limitations

The recruitment strategy relied heavily on social media, which may have introduced sampling bias by excluding individuals not active online. To mitigate this, a partnership with Endometriosis Australia facilitated wider outreach through their networks.

The study sample was predominantly composed of white Australians with higher education levels and middle-class incomes, limiting the findings’ applicability to migrants, minority ethnic groups, and Indigenous populations. This lack of diversity highlights gaps in our understanding of how class and race intersect with gender and ability, particularly concerning workplace policies (Acker, 2006 [51]). Further research focusing on underrepresented communities is essential to explore their unique experiences, address systemic inequities, and develop tailored workplace support.

It is unclear why some managers are supportive, inclusive, and empathetic to their team members and others are not. While it may be expected that female employers or managers may be more empathetic, this is not always the case. While female employees often feel more comfortable speaking to female managers, they can often be dismissive [62]. One possibility is because their own experiences of menstruation are less severe or less impactful and therefore they do not understand the range of experiences of those with endometriosis. Another explanation are the underlying norms of ‘menstrual etiquette’ where women encourage each other to conceal their symptoms associated with menstruation or otherwise face discreditation and social stigma at work [17,63]. Future work is required to better understand how to improve the understanding of the potential severity of symptoms and reduce stigma surrounding menstruation and endometriosis.

Despite these limitations, this study offers significant insights and expands the existing literature on improving workplace support for people with endometriosis.

## 5. Conclusions

Very little literature exists exploring employers’ perspectives of supporting employees with endometriosis at work. We sought to address this gap by understanding (RQ1) existing workplace policies and supports and attitudes towards proposed Endo@Work guidelines, as well as (RQ2) understanding how existing workplace interactions drive equitable workplace culture and support staff living and working with endometriosis.

The analysis of focus group discussions reveals that, while most workplaces outwardly value policy, guidelines and reasonable adjustments for employees with endometriosis, these commitments are often tokenistic, inconsistently applied, and inequitable. Theme 1 highlights these policy shortcomings, while Theme 2 explores the (banal) everyday practices—such as top-down leadership, trust, and advocacy—that drive cultural change towards equity.

To better support employees with endometriosis, we recommend that employers (1) implement tangible flexibility; (2) develop guidelines that integrate policy with education; (3) embed a dedicated advocacy role; and (4) adopt a top-down approach to secure leadership buy-in.

Our findings demonstrate that while employers acknowledge the importance of supporting employees with endometriosis, the findings highlight the need for consistent, equitable policies and practices, with recommendations emphasising flexibility, comprehensive guidelines, open and widespread communication, education to dispel stigma, and leadership engagement to foster workplace equity.

## Figures and Tables

**Table 1 healthcare-13-00930-t001:** Demographics, workplace, and role-related information (N = 24).

Gender, n (%)		
Women	22	92%
Men	2	8%
**Work in (Australian States & Territories) * n, %**		
Across all states and territories	8	33%
New South Wales	7	29%
Queensland	2	8%
South Australia	-	-
Tasmania	-	-
Victoria	10	42%
Western Australia	-	-
Australian Capital Territory	-	-
Northern Territory	1	4%
**Job Title, n (%)**		
Manager/Supervisor	13	54%
Executive Team	2	8%
HR Department	6	25%
Consultant	3	13%
**Industry/Field, n (%)**		
Healthcare and social assistance	4	17%
Education and training	2	8%
Administrative and support services	-	-
Professional, scientific, and technical services	4	17%
Financial and insurance services	4	17%
Retail trade	1	4%
Public administration and safety	-	-
Information media and telecommunications	2	8%
Construction	1	4%
Mining/manufacturing	2	8%
Other	-	-
Emergency services	2	8%
HR strategy	1	4%
Utilities	1	4%
**Number of Employees, n (%)**		
Up to 19	3	13%
20 to 49	2	8%
50 to 199	6	25%
Over 200	12	50%

* indicates n is greater than 24 as some participants’ workplaces spanned multiple States and/or Territories.

**Table 2 healthcare-13-00930-t002:** Themes and Subthemes.

Themes	Subthemes
“We have policies”	…but they are failing to fully support all staff …but they are inconsistent across managers …but they are only beneficial to the ideal worker
Workplace culture change is driven by the banal (everyday) practices	Change is driven from the top-down Change is driven by managers who trust staff and understand they are not homogenous Change is driven by a consistent education (and a role to facilitate it)

## Data Availability

The original contributions presented in this study are included in the article. Further inquiries can be directed to the corresponding author.

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
