# Peer review of "From Insight into Action: Understanding How Employer Perspectives Shape Endometriosis-Inclusive Workplace Policies"

_healthcare, 2025, doi:10.3390/healthcare13080930_

Round 1

Reviewer 1 Report

Comments and Suggestions for Authors
  • The topic of research is very important, as a large part of working society is affected by endometriosis. Not forgetting that this disease costs both companies and the economy money.

  • Employers' perceptions and assumptions regarding the management of employees with endometriosis were examined. Focus groups were formed for this purpose. However, most participating employers (92%) were female. 
    Since female employers may be affected by this or other gynecological diseases themselves, they may be more likely to recognize the need for organizational support for employees with chronic diseases such as endometriosis than male employers.
    This aspect should be given greater consideration. It could be mentioned in the “limitations” sector.

  • The topic of stigmatization is not addressed enough. People with illnesses can experience stigmatization in their everyday working lives. The various types of stigmatizations (e.g., employees feel they have to hide their chronic illness at work because they think others don't understand them, e.g., employees who already experience discrimination at work due to their chronic illness) should be addressed. Stigmatization should be mentioned again in the key recommendations. A key recommendation in this context could be a company-wide awareness campaign as a measure to counteract the stigmatization of people with endometriosis.

Reviewer 2 Report

Comments and Suggestions for Authors

This is a study which is a continuation of previous phases of a study to see the implementation of guidelines called Endo@Work. Authors also talk about EndoChampion which appears to be part of the Endo@Work guidelines.

While this is a good study and good information can come out of it is poorly written and needs to have major revision to make it readable and understandable.

Following are my specific comments/recommendations:

Introduction:  lines 33-41 is very strange I think the authors may have copied the authors’ instructions/guidelines  and forgotten to remove them. Introduction should start at line 41 with the sentence “As women's labour market participation..”. If there is a reason why authors have included the earlier segments, they should explain it

Line 47, authors write “as we will show” that should be determined only after the results are analysed. It is better if the authors just explain the existing body of knowledge and the gap they are trying to fill.

The footnote on page one is better explained in the main text of the introduction giving briefly the meaning of cis and trans gender to lay the ground for readers. In the data collection section, the explanation that this study is limited to cis gender can be further explained. The footnote would be better as part of the main introduction text, with reference.

Line 56  what is the difference between women and PFAB? At this point authors should explain that these two words can be used interchangeably and provide some reference to back their use of the words. The footnote is not helpful.

Line 67 women and those PFAB- sentence seems to be incomplete.

Line 89 why post pandemic was the problem not present before the pandemic? Or perhaps the previous phases were done before the pandemic if so that is not explained at all.

Introduction should be written better so that the foundation is laid for the study, a brief explanation of Endo@Work guidelines and any research undertaken with this guideline, who has developed the guideline? Brief history of the guideline and whether there were any other papers about implementation what were their challenges would also be helpful. What are the results of the earlier phases, is this study based on those results.

Introduction is not well written, authors should explain the terms women and PFAB and explain that it can be used interchangeably and the problem with endometriosis affects these groups of people.

In methods authors explain that this is the third phase of a wider project is anything published from the 1st 2 phases at all, if yes, they should include that in introduction as literature review. Were the participants in this phase also from the previous phases or were they completely new participants. And what is the relationship of this study with the previous phases?

How many discussions were held and how many participants in each discussion, were all the authors present in every group discussion? Were any participants included in more than one discussion. Were the discussions recorded?

Method section line 127 the 8% identifying as men are they PFAB which is not clear. The use of the word men here is a little bit confusing

Table 1 the numbers don't tally if n=24 some sections it's more than 24

Line 144 researchers staying online was it to discuss each focus group and the way in which the discussion was to be analysed?

Line 184 says that it compiled them alongside the authors reflection into word documents? It is not clear what the purpose of this was, is it reflections on every participant or every discussion, would this not dilute the responses from the participants.

Item 2.3 reviewing endo at work guidelines: based on what's written here there may be some information which can be included in the introduction session section to lead to the development of this study. This entire section can be part of the introduction.

Line 164 authors say that endo champion was discussed as one of the study themes where was it discussed, it seems to appear for the first time in this section if all this was explained in the introduction section it will be much easier for the readers to follow and to understand why this study is even being undertaken. What is the knowledge that the previous phases did not provide that this phase is attempting to provide. Was an EndoChampion included as a participant in this study?

Lines 188- and he197 is there any reference for this type of data analysis- what is NVivo, need to explain that it is a software, not all readers may know.

Line 199-206- how was this actually done the description given is a more theoretical description but how was it actually done in this study?

Results have to be better described what is theme one and theme two it is explained poorly  in the text but not in Table 2. Under the sub themes some are questions some are statements they should be consistent.

Authors talk about post pandemic was anything different pre pandemic and has a pandemic anything to do with the study at all?

Lines 224 what is Sam focus Group 1? Authors are introducing terms that have not been explained before and is not possible for readers to understand? Is Sam a participant if so, the name should not be provided.

The results section should have some charts to say what was the consensus of the participants if there are themes and sub themes how many people subscribed to which theme and subtheme? And then certain verbatim remarks can be used to emphasize the results of the group discussion. Charts can be used to show a percentage of the responses that led to identification of themes and subthemes.

Discussion line 455 how can the authors conclude that through his study findings a new normal was seen post pandemic when they did not have any pre pandemic comparison?

Line 502 Section 4.2 policy alone is not enough- this discussion can only be substantiated if the results are described better the discussion should analyze the results of the study to be able to come to some conclusions

Line 540 Section 4.3 key recommendation: there must be some mention of this in the discussion with substantiation from the results. Where in the results have the authors shown that these recommendations can be substantiated?

Reviewer 3 Report

Comments and Suggestions for Authors

The topic covered in the paper is very relevant and timely. The paper is written appropriately, well-structured, and provides valuable insights.
Nevertheless, I would like to make a few comments that may help the authors to improve the paper:

•    Remove Lines 33-41 from the introduction - these are the formal requirements for the introduction, given in the article template of Healthcare.
•    I would suggest removing the names of the informants. Since focus groups were used, mentioning the names allows for identifying the informant, especially within the focus group members.
•    At the end of sections 3.1.1-3.2.3, I suggest writing at least one sentence revealing the research's results. These sections now end with quotes from informants. The sentences at the beginning of the sections are general and do not show the research's results. However, the logic of qualitative research implies the necessity of generalization by the authors.

Round 2

Reviewer 2 Report

Comments and Suggestions for Authors

The paper is rewritten more clearly. Introduction explains the terms women, PFAB and gender diversities at the workplace. They have also explained in the introduction, use of terms like trans men, intersex etc and how menstruation affects this group.
They have explained the setting of the research to be post pandemic. They have also explained this study as the fourth phase and first three phases and what this fourth phase attempts to study.

Methods has been improved, and this revision provides a clear view of how the data was gathered. The PAR method is explained with more information.

The discrepancy in table 1 is explained

Reviewing endo at work guidelines is explained better and reads better. The term EndoChampion is explained and is important to this study as this individual will play a role in providing support and bringing change in the workplace as per the Endo@Work guidelines

Positioning Ourselves in Data Analyses Approach-  is modified to read better

Authors have added some conclusion after each result subheadings this gives a mini conclusion based on the responses. This helps to understand the discussion. Recommendations reflect the results and the discussion in this version.

Some errors in sentence construction is still present for example: Sentence incomplete: Strengthening workplace values which reproduce the “ideal worker” norm – an unencumbered employee who prioritises work above personal, familial, or social obligations [12,13,15]. There are other incomplete sentences that need to be corrected

Comments on the Quality of English Language

Some errors in sentence construction is still present for example: Sentence incomplete: Strengthening workplace values which reproduce the “ideal worker” norm – an unencumbered employee who prioritises work above personal, familial, or social obligations [12,13,15]. There are other incomplete sentences that need to be corrected
